# Contradiction Retrieval Via Sparse-Aware Sentence Embedding

## Abstract

Contradiction retrieval refers to identifying and extracting documents that explicitly disagree with or refute the content of a query, which is important to many downstream applications like fact checking and data cleaning. To retrieve contradiction argument to the query from large document corpora, existing methods such as similarity search and crossencoder models exhibit significant limitations. The former struggles to capture the essence of contradiction due to its inherent nature of favoring similarity, while the latter suffers from computational inefficiency, especially when the size of corpora is large. To address these challenges, we introduce a novel approach: SPARSECL that leverages specially trained sentence embeddings designed to preserve subtle, contradictory nuances between sentences. Our method utilizes a combined metric of cosine similarity and a sparsity function to efficiently identify and retrieve documents that contradict a given query. This approach dramatically enhances the speed of contradiction detection by reducing the need for exhaustive document comparisons to simple vector calculations. We validate our model using the Arguana dataset, a benchmark dataset specifically geared towards contradiction retrieval, as well as synthetic contradictions generated from the MSMARCO and HotpotQA datasets using GPT-4. Our experiments demonstrate the efficacy of our approach not only in contradiction retrieval with more than 30% accuracy improvements on MSMARCO and HotpotQA across different model architectures but also in applications such as cleaning corrupted corpora to restore high-quality QA retrieval. This paper outlines a promising direction for improving the accuracy and efficiency of contradiction retrieval in large-scale text corpora.

## 1 Introduction

Figure 1: Performance gains in NDCG@10 score across different sentence embedding models and datasets, showcasing the effectiveness and robustness of our SPARSECL compared with standard contrastive learning (CL)

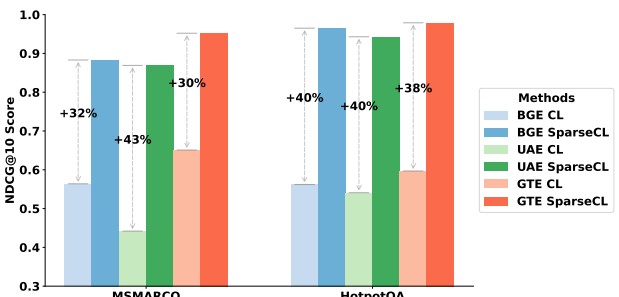

Training sentence embedding for similarity retrieval has been well studied in the literature (Gao et al. (2021); Xiong et al. (2020); Karpukhin et al. (2020)), where a standard practice is to use contrastive learning to map those similar sentences together and those dissimilar sentences far from each other. However, these existing sentence embeddings are mainly tailored to similarity retrieval, while as far as we know, there hasn't been sentence embeddings for non-simlarity based retrieval. In this paper, we study the problem of contradiction retrieval, a typical case of non-similarity based retrieval. Given a large document corpus and a query passage, the goal is to retrieve document(s) in the corpus that contradict the query, assuming they exist. This problem has a large number of applications, including counter-argument detection Wachsmuth et al. (2018) and fact verification Thorne et al. (2018). The standard approaches to retrieving contradictions are two-fold. One is to use a bi-encoder Xiao et al.

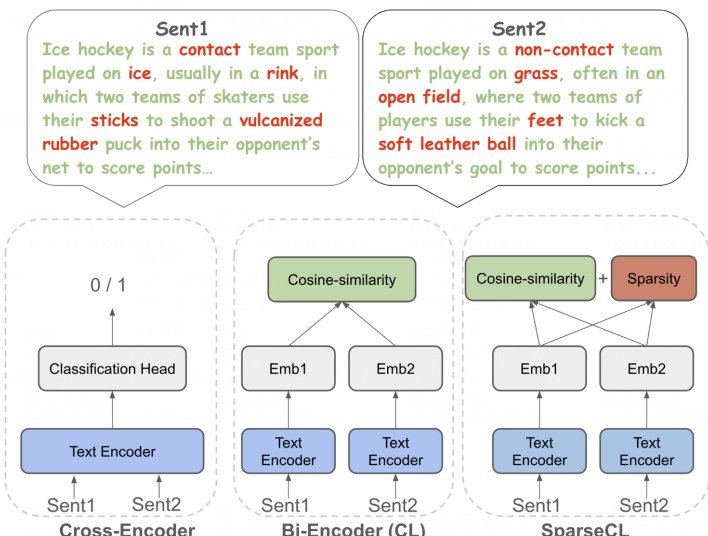

Figure 2: Comparison of our SPARSECL with Cross-Encoder and Contrastive-Learning based Bi-Encoder for contradiction retrieval.

(2023); Li & Li (2023); Li et al. (2023) that maps each document to a feature space such that two contradicting documents are mapped close to each other (e.g., according to the cosine metric) and use nearest neighbor search algorithms. The second approach is to train a cross-encoder model Xiao et al. (2023) that determines whether two documents contradict each other, and apply it to each document or passage in the corpus.

Unfortunately, both methods suffer from limitations. The first approach (cosine similarity search on sentence embeddings) is inherently incapable of representing the "contradiction relation" between the documents, due to the fact that the cosine metric is transitive: if $A$ is similar to $B$, and $B$ is similar to $C$, then $A$ is also similar to $C$. As an example, consider an original sentence and its paraphrase in Table 8. Both of them contradict the sentence in the third column but they are not contradicting each other. The second approach, which uses a cross-encoder model, can capture the contradiction between sentences to some extent, but it is much more computationally expensive. Our experiment in Appendix H shows that compared with standard vector computation, running a cross-encoder is at least 200 times slower.

In this paper, we propose to overcome these limitations by introducing SPARSECL for efficient contradiction retrieval using sparse-aware sentence embeddings. The key idea behind our approach is to train a sentence embedding model to preserve *sparsity of differences* between the contradicted sentence embeddings. When answering a query, we calculate a score between the query and each document in the corpus, based on *both* the cosine similarity and the sparsity of the difference between their embeddings, and retrieve the ones with the highest scores. Our specific measure of sparsity is defined by the Hoyer measure of sparsity Hurley & Rickard (2009), which uses the scaled ratio of the $\ell_1$ norm and the $\ell_2$ norm of a vector as a proxy of the number of non-zero entries in the vector. Unlike the cosine metric, the Hoyer measure is not transitive (please refer to Appendix D for a detailed analysis), which avoids the limitations of the former. At the same time this method is much more efficient than a cross-encoder, as both the cosine metric and the Hoyer measure are easy to compute given the embeddings. The Hoyer sparsity histogram of our trained embeddings is displayed in Figure 3.

We first evaluate our method on the counter-argument detection dataset Arguana Wachsmuth et al. (2018), which to the best of our knowledge, is the only publicly available dataset suitable for testing contradiction retrieval. In addition, we generate two synthetic data sets, where contradictions for documents in MSMARCO Nguyen et al. (2016) and HotpotQA Yang et al. (2018) datasets are synthetically generated using GPT-4 Achiam et al. (2023). Our experiments demonstrate the efficacy of our approach in contradiction retrieval, as seen in Table 1. We also apply our method to corrupted corpus cleaning problem, where the goal is to filter out contradictory sentences in a corrupted corpus and preserve good QA retrieval accuracy.

To summarize. our contributions can be divided into three folds:

- We introduce a novel contradiction retrieval method that employs specially trained sentence embeddings combined with a metric that includes both cosine similarity and the Hoyer measure of sparsity. This approach effectively captures the essence of contradiction while being computationally efficient.

- Our method demonstrates superior performance on both real and synthetic datasets, achieving significant improvements in contradiction retrieval metrics compared to existing methods. This underscores the effectiveness of our embedding and scoring approach.

- We apply our contradiction retrieval method to the problem of corpus cleaning, showcasing its utility in removing contradictions from corrupted datasets to maintain high-quality QA retrieval. This application highlights the practical benefits of our approach in real-world scenarios.

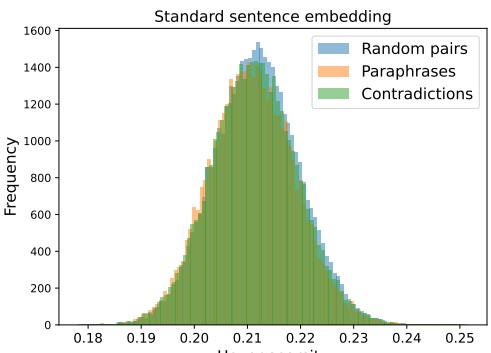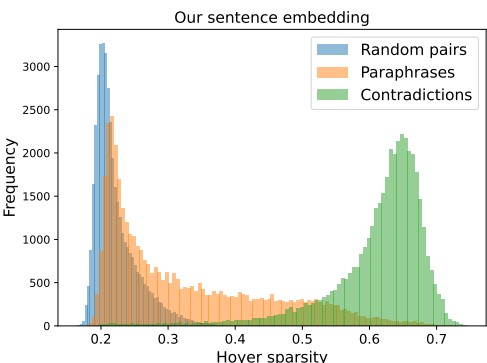

Figure 3: Histograms for the Hoyer sparsity of different pairs of sentence embedding differences on HotpotQA test set. The left figure is the histogram produced by a standard sentence embedding model ("bge-base-en-v1.5"), where the median Hoyer sparsity values for random pairs, paraphrases, and contradictions are $0.212, 0.211, 0.211$. The right figure is the histogram produced by our sentence embedding model fine-tuned from "bge-base-en-v1.5" using our SPARSECL method, where the median Hoyer sparsity values for random pairs, paraphrases, and contradictions are $0.212, 0.281, 0.632$.

## 2 RELATED WORK

**Counter Argument Retrieval**  A direct application of our contradiction retrieval task in "counter-argument retrieval". Since the curation of Arguana dataset by Wachsmuth et al. (2018), there has been a few previous work on retrieving the best counter-argument for a given argument Orbach et al. (2020); Shi et al. (2023). In terms of methods, Wachsmuth et al. (2018) uses a weighted sum of different word and embedding similarities and Shi et al. (2023) designs a "Bipolar-encoder" and a classification head. We believe that our method relying only on cosine similarity and sparsity is simpler than theirs and produces better results in the experiment. In addition, some analyses in the counter-argument retrieval papers are specific to the "debate" setting, e.g. they rely on topic, stance, premise/conclusion, and some other inherent structures in debates for help, which may prevent their methods from being generalized to broader scenarios.

**Fact verification and LLM hallucination**  Addressing the hallucination problem in Large Language Models has been a subject of many research efforts in recent years. According to the three types of different hallucinations in Zhang et al. (2023b), here we only focus on those so called "Fact-Conflicting Hallucination" where the outputs of LLM contradict real world knowledge. The most straightforward way to mitigate this hallucination issue is to assume an external groundtruth knowledge source and augment LLM's outputs with an information retrieval system. There have been a few works on this line showing the success of this method Ren et al. (2023); Mialon et al. (2023). This practice is very similar to "Fact-Verification" Thorne et al. (2018); Schuster et al. (2021) where the task is to judge whether a claim is true or false based on a given knowledge base.

However, as pointed out by Zhang et al. (2023b), in the era of LLM, the external knowledge base can encompass the whole internet. It is impossible to assume that all the information there are perfectly correct and there may exist conflicting information within the database. In the context of our paper,

instead of using a groundtruth database to check an external claim, our goal is to check the internal contradictions between different documents in an unknown corpus.

**Learning augmented LLM and retrieval corpus attack**    Augmenting large language models with retrieval has been shown to be useful for many purposes. Recently, there have been a few works Zhong et al. (2023); Zou et al. (2024) studying the vulnerability of retrieval system from adversarial attack. In specific, they show that adding a few corrupted data to the corpus will significantly drop the retrieval accuracy. This phenomenon bring our attention to the necessity of checking the factuality of the knowledge database. Note that the type of corrupted documents considered by their papers are different from ours. While they consider the injection of adversarially generated documents, we consider the existence of contradicted documents as a natural part of the corpus. Also their purpose is to show the effect of adversarial attack, while we provide a defense method for a certain kind of corrupted database.

## 3    METHOD

**Problem Formulation**    We consider the contradiction retrieval problem: given a passage corpus $C = \{p_1, p_2, ...p_n\}$ and a query passage $q$, retrieve the "best" passage $p^*$ that contradicts $q$. We assume that several similar passages supporting $q$ might exist in the corpus $C$.

**Embedding based method**    Judging whether two passages contradict each other is a standard Natural Language Inference task and can be easily tackled by many off-the-shelf language models Touvron et al. (2023); Xu et al. (2022), . However, to retrieve the best candidate from the corpus, we have to iterate the whole corpus, or at least send the candidates retrieved by similarity search to the language model to determine if they constitute contradiction. This is time consuming, given that there are potentially many similar passages in the corpus. Therefore, in our paper, we mainly focus on those methods that only rely on their passage embeddings. Specifically, we want to design a simple scoring function $F$ that given the embeddings of two passages, outputs a score between $[0, 1]$, indicating the likelihood that they are contradicting each other.

**Sparse Aware Embeddings**    Following the idea from counter-argument retrieval papers Wachsmuth et al. (2018), such a score function should be a combination of similarity and dissimilarity functions. Observe that a dissimilarity function is basically a negation of a similarity function, so the authors of Wachsmuth et al. (2018) design several different similarity functions and set the scoring function to maximize one of them and minimize another. Here, instead of enumerating different similarity functions, we consider another notion: the "sparsity" of their embedding differences. The basic intuition is as follows. Suppose that all sentences are represented as vectors in a "semantic" basis, where each coordinate represents one clearly identifiable semantic meaning. Then a contradiction between two passages should manifest itself as a difference in a few coordinates, while other coordinates should be quite close to each other. The issue, however, is that we do not know how to construct the appropriate basis, and the sparsity is defined with respect to a fixed coordinate system. Nevertheless, following this intuition, we fine-tune sentence embedding models using contrastive learning, by rewarding the sparsity of the difference vectors between embeddings of contradicting passages. Please see Figure 3 for the Hoyer sparsity histogram of our trained embeddings.

**SPARSECL**    We use contrastive learning (Gao et al. (2021); Karpukhin et al. (2020)) to fine-tune any pretrained sentence embedding model to generate the desired sparsity-aware embeddings. The choice of positive and negative examples are exactly the reverse of the choice we make when the training sets are Natural Language Inference datasets. The positive example for a passage is its contradiction passage in the training set. The hard negative example for a passage is its similar passage in the training set. There are also other random in-batch passages as soft negative examples. The sparsity function we choose here is Hoyer sparsity function from Hurley & Rickard (2009). Let $h_1$ and $h_2$ be two sentence embeddings and their embeddings have dimension $d$. We define

$$\text{Hoyer}(h_1, h_2) = \left( \sqrt{d} - \frac{\|h_1 - h_2\|_1}{\|h_1 - h_2\|_2} \right) / \left( \sqrt{d} - 1 \right).$$

This is a transformed version of the ratio of the $l_1$ to the $l_2$ norm, with output normalized to $[0, 1]$.

Finally, for each training tuple $(x_i, x_i^+, x_i^-)$ with their embeddings $(h_i, h_i^+, h_i^-)$, batch size $N$, and temperature $\tau$, its loss function is defined as

$$l_i = -\log \frac{e^{\text{Hoyer}(h_i, h_i^+)/\tau}}{\sum_{j=1}^{N} \left( e^{\text{Hoyer}(h_i, h_j^+)/\tau} + e^{\text{Hoyer}(h_i, h_j^-)/\tau} \right)}.$$

**Scoring function for contradiction retrieval** For the score function for contradiction retrieval, we use a weighted sum of the standard cosine similarity and our sparsity function. Note that the cosine similarity is provided separately by any off-the-shelf sentence embedding model in a zeroshot manner. It can can also be fine-tuned. Let $E()$ be the standard sentence embedding model and $E_s()$ be our sparse-aware sentence embedding model trained by SPARSECL. Then the final score function for contradiction retrieval is

$$F(p_1, p_2) = \cos \left( E(p_1), E(p_2) \right) + \alpha \cdot \text{Hoyer}(E_s(p_1), E_s(p_2)).$$

where $\alpha$ is a scalar tuned using the validation set. Note that the criterion for contradiction is usually case-dependent, so it is necessary that we reserve a parameter to adapt to different notions of contradiction. To get the answer passages, we calculate the score function for all passages and report the top 10 of them[1].

## 4 EXPERIMENTS

We test our contradiction retrieval method on a counterargument retrieval task Arguana Wachsmuth et al. (2018) and two synthetic datasets adapted from HotpotQA Yang et al. (2018) and MS-MARCO Nguyen et al. (2016). Then, we apply our contradiction retrieval task to a new experimental setting: retrieval corpus cleaning. Finally, we perform ablation studies to explain the functionality of each component of our method. Most of our experiments are not so computationally extensive, which can be run by one single A6000 GPU. We run our major experiments on A6000 and A100 GPUs.

### 4.1 COUNTER-ARGUMENT RETRIEVAL

**Dataset** Arguana is a dataset curated in Wachsmuth et al. (2018), where the author provide a corpus of 6753 argument-counterargument pairs, taken from 1069 debates with 15 themes on idebate.org. For each debate, the arguments are further divided into two opposing stances (pro and con). For each stance, there are paired arguments and counter-arguments. The dataset is split into the training set (60% of the data), the validation set (20%), and the test set (20%). This ensures that data from each individual debate is included in only one set and that debates from every theme are represented in every set. The task goal is: given an argument, retrieve its best counter-argument.

**Training** We use Arguana's training set to fine-tune our sparsity aware sentence embedding model via SPARSECL. To construct our training data, for each argument and counter-argument pair $(x_i, x_i^c)$ in the Arguana's training set, we set $x_i^c$ to be the positive example of $x_i$. We select all the other arguments and counter-arguments from the same debate and stance as $x_i$'s hard negatives. We fine-tune three pretrained sentence embedding models of different sizes ("UAE-Large-V1" Li & Li (2023), "GTE-large-en-v1.5" Li et al. (2023), and "bge-base-en-v1.5" Xiao et al. (2023)). Please refer to Table 12 for our training parameters.

**Baselines** We mainly compare our method to the similarity-based method. Since Arguana is one of the datasets in the MTEB Retrieval benchmark, directly searching for the similar passages in the corpus can already produce quite good test results. We report the performance of several efficient (with fewer than 1B parameters) and top-ranked pretrained sentence embedding models including "GTE-large-en-v1.5", "UAE-Large-V1", "bge-base-en-v1.5", when used to directly retrieve the most similar argument to each query (Zeroshot). For a fair comparison, we also report the results of fine-tuning these models using standard contrastive learning (CL) on the same dataset used for SPARSECL.

---

[1]In the actual implementation, for time efficiency, we first use FAISS Douze et al. (2024) to retrieve the top K candidates with cosine similarity and then rerank them using our cosine + sparsity score function. We set a very large $K$ (e.g. $K = 1000$) so that empirically this is almost equivalent to searching for the maximal cosine + sparsity score in the whole corpus

| Model | Method | Arguana | MSMARCO | HotpotQA |
|-------|--------|---------|---------|----------|
| BGE | Zeroshot (Cosine) | 0.658 | 0.600 | 0.595 |
|     | **Zeroshot (Cosine) + SPARSECL(Hoyer)** | **0.704** | **0.909** | **0.967** |
| BGE | CL (Cosine) | 0.687 | 0.527 | 0.562 |
|     | **CL (Cosine) + SPARSECL(Hoyer)** | **0.722** | **0.883** | **0.965** |
| UAE | Zeroshot (Cosine) | 0.683 | 0.597 | 0.587 |
|     | **Zeroshot (Cosine) + SPARSECL(Hoyer)** | **0.743** | **0.902** | **0.955** |
| UAE | CL (Cosine) | 0.704 | 0.442 | 0.541 |
|     | **CL (Cosine) + SPARSECL(Hoyer)** | **0.744** | **0.869** | **0.943** |
| GTE | Zeroshot (Cosine) | 0.725 | 0.603 | 0.597 |
|     | **Zeroshot (Cosine) + SPARSECL(Hoyer)** | **0.797** | **0.953** | **0.977** |
| GTE | CL (Cosine) | 0.778 | 0.651 | 0.597 |
|     | **CL (Cosine) + SPARSECL(Hoyer)** | **0.813** | **0.952** | **0.979** |

Table 1: Results for different models and methods on the contradiction retrieval task. Experiments are run on the Arguana dataset Wachsmuth et al. (2018) and modified MSMARCONguyen et al. (2016) and HotpotQAYang et al. (2018) datasets. We report NDCG@10 score here, the higher the better. "UAE" stands for "UAE-Large-V1", "BGE" stands for "bge-base-en-v1.5", "GTE" stands for "gte-large-en-v1.5", The "Method" column denotes the score function used to retrieve contradictions. We consider two score functions: cosine similarity and cosine similarity plus Hoyer sparsity. "Zeroshot" denotes the direct testing of the model without any fine-tuning. "CL" denotes fine-tuning using standard contrastive learning. "SPARSECL" denotes fine-tuning using Hoyer sparsity contrastive learning (our method).

**Test**  The Arguana test set consists of 1401 query arguments and counter-argument pairs. Following the standard test setting, we search for an answer of a query within the whole corpus (training set + validation set + test set) and report NDCG@10 scores. The $\alpha$ parameter we used in the score function varies across different datasets and models. We select $\alpha$ based on the best NDCG@10 score on the validation set. Please refer to Table 13 in Appendix G for our specific $\alpha$ choices and parameter searching details. When we directly use a model to provide cosine similarity scores in a zeroshot manner, we use its default pooler ("cls") for that model. When we use a fine-tuned model (via either CL or SPARSECL) to provide either cosine similarity scores or sparsity scores, we use the "avg" pooler.

**Results**  The detailed results are presented in Table 1. Across all models—"GTE-large-en-v1.5", "UAE-Large-V1", and "bge-base-en-v1.5"—an average improvement of 4.8% in counter-argument retrieval were observed when incorporating our SPARSECL to either Zeroshot or CL. Furthermore, our CL (Cosine) + SPARSECL (Hoyer) method achieves NDCG@10 score 0.813 using GTE with only 400M parameters. For completeness, we also compare our results with Shi et al. (2023) in Appendix E.

This pattern of enhancement was consistently observed regardless of whether the embedding models were fine-tuned or not. Notably, standard cosine similarity fine-tuning alone also contributed to performance gains. For instance, fine-tuned GTE models showed an increase from 0.725 to 0.778 on the Arguana dataset using standard cosine similarity alone. This suggests that the Arguana dataset inherently favors scenarios where the counterargument is the most similar passage to the query, which may amplify the benefits of fine-tuning.

These findings highlight the robustness of our approach, particularly when traditional similarity metrics are augmented with sparsity measures to capture subtle nuances in contradiction. Further insights can be gleaned from our ablation study detailed in Section 4.5, where we analyze the impact of similar non-contradictory passages within the corpus.

## 4.2 CONTRADICTION RETRIEVAL ON SYNTHETIC DATASETS

The task of "contradiction retrieval" generalizes beyond the argument and counter-argument relationship in the debate area, e.g. passages with conflicting factual information should also be considered

as "contradictions". To test our method's validity for these more general forms of contradictions, we construct two synthetic datasets to test our method's performance.

**Data set construction** Given a QA retrieval dataset, e.g. MSMARCO Nguyen et al. (2016), for each answer passage $x_i$ of a query $q_i$, we use Large Language Models (specifically, GPT-4 Achiam et al. (2023)) to generate 3 synthetic answers paraphrasing $x_i$ or contradicting $x_i$. Let the generated paraphrases be $\{x_{i1}^+, x_{i2}^+, x_{i3}^+\}$ and the generated contradictions be $\{x_{i1}^-, x_{i2}^-, x_{i3}^-, \}$. We then delete $x_i$ from the corpus and add the set of generated passages $\{x_{i1}^+, x_{i2}^+, x_{i3}^+, x_{i1}^-, x_{i2}^-, x_{i3}^-\}$ to the corpus. In the test phrase, the queries are $\{x_{i1}^+, x_{i2}^+, x_{i3}^+\}$, each of which has the same answers $\{x_{i1}^-, x_{i2}^-, x_{i3}^-, \}$. We generate the paraphrases and contradictions for the validation set, test set, and a randomly sampled 10000 documents from the training set.

The reason why we only keep the generated text but not the original one is that all the GPT-4 generated passages are easily distinguishable from the human written ones, which makes language models vulnerable to shortcuts. Please refer to Table 8 to see two examples of the generated paraphrases and contradictions. We report the prompts and the temperature parameter we use to generate these data in Appendix B.

**Training** To prepare the training data for contrastive learning, for each paraphrase and contradiction set $\{x_{i1}^+, x_{i2}^+, x_{i3}^+, x_{i1}^-, x_{i2}^-, x_{i3}^-\}$ generated from the same original passage, we form 9 pieces of training data $(x_{ia}^+, x_{ib}^-, x_{ic}^+)$ for 9 different combinations of paraphrases, contradictions, and a randomly selected hard negative from the remaining two paraphrases. We then perform SPARSECL to fine-tune a sparsity-enhanced embedding.

**Baseline** We are not aware of any accurate methods for retrieving contradictions that only rely on sentence embeddings. Therefore, the only baseline we provide is a standard contrastive learning with cosine similarity (CL), using the same training data (contradictions as positive examples and paraphrases as negative examples) that we use for our SPARSECL.

**Test** Similar to the testing strategy for Arguana, we define our corpus to consist of all generated text (training set + validation set + test set). We query the paraphrases $\{x_{i1}^+, x_{i2}^+, x_{i3}^+\}$ of the original passage $x_i$ and set the groundtruth answers to be the generated contradictions $\{x_{i1}^-, x_{i2}^-, x_{i3}^-\}$. We select the $\alpha$ parameter with the maximal NDCG@10 score on the validation set and report the NDCG@10 score obtained by applying that $\alpha$ to the test set.

The results are reported in Table 1. For both MSMARCO and HotpotQA data sets, incorporating our SPARSECL method achieves over 30 percentage points gain compared with the pure cosine-similarity-based method. The large improvement is due to the existence of paraphrases in the corpus, that are strong confounders for the pure similarity-based methods. We also observe that fine-tuning using standard contrastive learning with cosine similarity (CL) yields performance gains for Arguana but not for MSMARCO and HotpotQA. Our explanation is that, for MSMARCO and HotpotQA, the generated paraphrases are more similar to the query than the contradictions. Therefore fine-tuning with the standard cosine similarity is unlikely to work.

### 4.3 ZERO-SHOT GENERALIZATION TEST

To evaluate the generalization capability of our sparse-aware embeddings, we also conduct zero-shot tests on other datasets. Specifically, we train the embeddings on our synthetic HotpotQA or MSMARCO datasets and then test them on the other dataset in a zero-shot manner. As presented in Table 2, SparseCL trained on MSMARCO or HotpotQA produces reasonable test results on the other dataset, albeit with a slight performance drop. This demonstrates that the sparse-aware embeddings trained on one dataset can capture contradiction relationships and generalize to unseen datasets.

### 4.4 RETRIEVAL CORPUS CLEANING

As an application of contradiction retrieval, we test how well our method can be used to find inconsistencies within a corpus and clean the corpus for future training or QA retrieval. We first inject corrupted data contradicting existing documents into the corpus, and measure the retrieval

| Model | Method | Train Dataset | Test Dataset | NDCG@10 |
|-------|--------|---------------|--------------|---------|
| BGE | Zeroshot(Cosine)+SparseCL(Hoyer) | MSMARCO HotpotQA | HotpotQA MSMARCO | 0.886 0.877 |
| BGE | Zeroshot(Cosine)+SparseCL(Hoyer) | HotpotQA MSMARCO | HotpotQA MSMARCO | 0.967 0.909 |
| BGE | Zeroshot(Cosine) | N/A N/A | HotpotQA MSMARCO | 0.595 0.600 |

Table 2: Results for zero-shot generalization experiment for contradiction retrieval

accuracy degradation for retrieved answers. Then, we use our contradiction retrieval method to filter out corrupted data and measure the retrieval accuracy again.

**Data**  Similarly to the data generation in Section 4.2, we construct a new corpus containing LLM-generated paraphrases and contradictions based on MSMARCO and HotpotQA data sets. We start with an original corpus $C$ and its subset $S$. We then generate paraphrases and contradictions for $S$ as in Section 4.2.

For HotpotQA, $S$ contains all answer documents for the test set, $10000$ answer documents sampled from the training set, and $1000$ answer documents sampled from the development set. For MS-MARCO, $S$ contains all answer documents for the dev set, and $11000$ answer documents sampled from the training set.

We then curate 3 different versions of the corpus based on the original corpus $C$ and the subset $S$.

- The initial corpus $C^+$: For each original answer document $x$ in $S$, we remove $x$ from $C$ and instead add 3 LLM-generated paraphrases $\{x_1^+, x_2^+, x_3^+\}$ to $C$. The result forms the *initial* corpus $C^+$.
- The corrupted corpus $C^-$: For each original answer document $x$ in $S$, we generate 3 contradictions $\{x_1^-, x_2^-, x_3^-\}$ and add them to $C^+$ to get the *corrupted* corpus $C^-$.
- The cleaned corpus $C^\natural$: We apply our data cleaning procedure to the corrupted corpus $C^-$, obtaining the *cleaned* dataset $C^\natural$.

**Test**  We test the retrieval accuracy (NDCG@10) and the corruption ratio (Recall@10) for answering the original queries in the test set. The goal of our experiment is to show how retrieval algorithms behave on these three constructed corpora $C^+$, $C^-$, and $C^\natural$.

**Data Cleaning**  Our sparsity-based method can only identify contradictions within the data set, but we do not know which element in a contradiction pair is correct. To perform data cleaning, we make the assumption that for each original passage $x \in S$, we are given one of its paraphrases as the groundtruth. Then, our task is reduced to searching for passages contradicting a given ground truth document and filtering them out.

**Method**  We use the GTE-large-en-v1.5 model without fine-tuning to provide the cosine similarity score for this data cleaning experiment. We use the model from our contradiction retrieval experiment in section 4.2 trained on MSMARCO and HotpotQA to provide the sparsity score. The $\alpha$ parameter is also identical to the one used in section 4.2. For each ground truth document, we filter out the top 3 scored documents from the corpus.

Note that the optimal choice of $\alpha$ for contradiction retrieval may not be the optimal choice for data cleaning because of different test objectives. We apply the same $\alpha$ only for simplicity, as our goal is to demonstrate the validity of applying our method to the data cleaning problem.

Table 3 shows the results. We observe that the retrieval accuracy on the corrupted corpus drops significantly, as the generated contradictions cause the embedding model to retrieve them as query answers. The corruption ratio measures the average fraction of the top-10 retrieved documents that correspond to the generated contradicting passages. This performance is above 40% for both datasets.

| Datasets | Original Acc | Corrupted Acc | Corrupt | Cleaned Acc | Corrupt |
|---|---|---|---|---|---|
| HotpotQA | 0.676 | 0.567 | 0.443 | 0.652 | 0.020 |
| MSMARCO | 0.435 | 0.381 | 0.413 | 0.414 | 0.040 |

Table 3: Experimental results for the impact of corrupted data on QA retrieval and contradiction retrieval for filtration. "Acc" represents the retrieval accuracy measured by the NDCG@10 score and "Corrupt" represents the fraction of returned passages that are corrupted, as measured by Recall@10.

After performing our corpus cleaning procedure, which searches for the passages contradicting the given ground truth documents and removes the top-3 for each of them, we can recover more than 60% of the performance loss due to corruption and at the same time reduce the corruption ratio to less than 5%.

## 4.5 ABLATION STUDIES

We perform the following three ablation studies to further understand sparsity-based retrieval method.

**Arguana retrieval results analysis** In the standard Arguana dataset, even though the task is to retrieve the counter-argument for the query, the retrieval based solely on similarity still gives reasonable results. This means that counter-arguments are also the most similar arguments to the query, which makes the data set an imperfect test bed for testing contradiction retrieval.

To further compare our sparsity-based method and the pure similarity-based method , we augment Arguana by adding arguments' paraphrases to the corpus. Specifically, for any argument $x$ and its counter-argument $x^-$ in the original corpus $C$, we use GPT-4 to generate three paraphrases $\{x_1, x_2, x_3\}$ of $x$. We then form three new corpora with an increasing number of paraphrases added to the corpus: $C_1$ contains all $x_1$ and $x^-$, $C_2$ contains all $x_1, x_2$, and $x^-$, and $C_3$ contains all $x_1, x_2, x_3$, and $x^-$.

In the testing phase, we query the counter-arguments for one of $x$'s paraphrases, the answer of which should still be $x^-$. We observe how the performance varies when the corpora we retrieve from are $C_1, C_2, C_3$.

| Models | Methods | $C_1$ | $C_2$ | $C_3$ |
|---|---|---|---|---|
| BGE | Zeroshot (Cosine) | 0.561 | 0.355 | 0.267 |
| | **Zeroshot (Cosine) + SPARSECL(Hoyer)** | 0.682 | 0.679 | 0.675 |
| BGE | CL (Cosine) | 0.471 | 0.303 | 0.228 |
| | **CL (Cosine) + SPARSECL(Hoyer)** | 0.619 | 0.618 | 0.615 |

Table 4: Counter-argument retrieval results on the augmented Arguana dataset with different numbers of similar arguments in the corpus. $C_x$ denotes testing counter-argument retrieval on the corpus with $x$ existing paraphrases (including itself) of the query argument.

We present our overall experimental results in Table 4. Please also refer to Appendix F for an example case study. As the number of paraphrases in corpus increases from 1 to 3, the performance of the similarity-based method drops significantly. Thus it is reasonable to deduce that, as the number of similar arguments in the corpus increases further, the NDCG@10 scores for similarity-based methods will converge to 0. On the other hand, the performance of our sparsity-based method is stable with respect to the number of paraphrases in the corpus.

**Different Scoring function for contradiction retrieval** We experiment with 5 other retrieval methods in our ablation study. The methods evaluated are as follows: "Prompt" involves appending the "Not true: " prompt to the query during testing, followed by standard similarity search. "Prompt + CL (Cosine)" extends this by incorporating contrastive learning with the "Not true: " prompt included in the training data. "Gen" uses GPT-4 to generate contradictions to the query (details in Appendix B) and applies similarity search for testing. "Gen + CL (Cosine)" fine-tunes using contrastive learning

| Model | Method | Arguana |
|-------|--------|---------|
| BGE | Prompt + Zeroshot (Cosine) | 0.657 |
|     | Gen + Zeroshot (Cosine) | 0.647 |
|     | Zeroshot (Cosine) | 0.658 |
| BGE | Prompt + CL (Cosine) | 0.645 |
|     | Gen + CL (Cosine) | 0.700 |
|     | CL (Cosine) | 0.687 |
| BGE | SparseCL (Hoyer) | 0.561 |
|     | CL (Cosine) + SparseCL (Hoyer) | 0.722 |

Table 5: Counter-argument retrieval results (NDCG@10 scores) on Arguana dataset with different retrieval methods. "Gen" means using GPT-4 to generate a contradiction $c$ of the query argument $q$, "Prompt" means appending the "Not true : " prompt in the front of the query text. "Zeroshot" refers to direct testing and "CL" and "SparseCL" refer to fine-tuning with respective methods.

with the generated contradictions in the training data before similarity search. Finally, "SparseCL (Hoyer)" employs SparseCL fine-tuning and retrieves documents based on the maximal Hoyer sparsity score during testing.

As shown in Table 5, we observe that generally "Gen" and "Prompt" don't improve much upon standard similarity search. For the "Gen + CL (Cosine)" method, a diverse set of counter-arguments exist for a given argument, making it hard to generate a single counter-argument that closely matches the true ground truth counter-argument. For the "Prompt + CL (Cosine)" method, fine-tuning with the appended prompt even results in a performance drop. During the training process, we observed overfitting and hypothesize that the special prompt "Not true:" introduces a shortcut, making it easier for the model to learn whether a text belongs to the "argument" class or the "counter-argument" class. However, this class information is not useful when identifying pairwise contradiction relationships. Finally, directly using Hoyer sparsity to retrieve contradictions doesn't yield good results as well, because we believe contradictions involve a combination of similarity and dissimilarity.

**Different sparsity functions** Our intuition in Section 3 does not give clear guidelines on which sparsity function to use in our SPARSECL. Thus, we also experiment with different choices of sparsity functions, selected from Hurley & Rickard (2009). Specifically, we consider two other sparsity functions ($l_2/l_1$ and $\kappa_4$), which are scale invariant and differentiable (see Table III in Hurley & Rickard (2009)). Note that both of these two sparsity functions have ranges $[0, 1]$, and higher values of those functions correspond to sparser vectors.

$$\frac{l_2}{l_1} = \frac{\|h_1 - h_2\|_2}{\|h_1 - h_2\|_1} \qquad \kappa_4 = \frac{\|h_1 - h_2\|_4^4}{\|h_1 - h_2\|_2^2}.$$

| Model | Method | $l_2/l_1$ | $\kappa_4$ | Hoyer | Cosine (baseline) |
|-------|--------|-----------|------------|-------|-------------------|
| BGE | Zeroshot (Cosine) + SPARSECL | 0.675 | 0.684 | **0.704** | 0.657 |
| BGE | CL (Cosine) + SPARSECL | 0.702 | 0.707 | **0.722** | 0.687 |

Table 6: NDCG@10 scores for Arguana using SPARSECL with different sparsity functions. We also report two baselines that use only the cosine similarity (zeroshot and contrastive learning).

As per Table 6, compared to the cosine similarity method, the combination of the cosine similarity score with the sparsity score trained by SPARSECL, yields higher NDCG@10 scores for each sparsity function. However, Hoyer sparsity yields the highest accuracy. We believe that simple sparsity functions have a more benign optimization landscape and thus are easier for models to optimize.

## 5 CONCLUSION

In this work, we introduced a novel approach to contradiction retrieval that leverages sparsity-aware sentence embeddings combined with cosine similarity to efficiently identify contradictions in large document corpora. This method addresses the limitations of the traditional similarity search as well as computational inefficiencies of the cross-encoder models, proving its effectiveness on benchmark datasets like Arguana and on synthetic contradictions retrieval from MSMARCO and HotpotQA.

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

## A    SCORE FUNCTIONS FOR NATURAL LANGUAGE INFERENCE TASK

As an application of our SPARSECL method, we demonstrate that our method can be useful for distinguishing entailments and contradictions in natural language inference datasets. For SNLI Bowman et al. (2015) and MNLI Williams et al. (2018) datasets, we extract entailment and contradiction pairs, fine-tune using standard contrastive learning and our SPARSECL, and then report the average cosine similarity / Hoyer sparsity score between entailments, contradictions, and random pairs.

|  |  | Contradiction | Entailment | Random |
|---|---|---|---|---|
| SNLI | Zeroshot (Cosine) | 0.546 | 0.769 | 0.376 |
|  | CL (Cosine) | 0.885 | 0.886 | 0.777 |
|  | SparseCL (Hoyer) | 0.376 | 0.347 | 0.228 |
| MNLI | Zeroshot (Cosine) | 0.659 | 0.818 | 0.378 |
|  | CL (Cosine) | 0.919 | 0.917 | 0.733 |
|  | SparseCL (Hoyer) | 0.422 | 0.364 | 0.244 |

Table 7: Average Cosine / Hoyer scores between Contradiction / Entailment / Random pairs of texts. The experiment is run on "bge-base-en-v1.5" model. Texts pairs are from SNLI and MNLI datasets

We can observe from Table 7 that, in the zeroshot setting, the average cosine similarity of contradiction pairs lies between the ranges of random and entailment pairs. For the fine-tuned model using standard contrastive learning (CL), the average cosine similarity of contradiction pairs is almost indistinguishable from that of entailment pairs. Finally, after being fine-tuned using SPARSECL, the model exhibits higher average Hoyer sparsity scores for contradiction pairs compared to other two types of relationships.

## B    DATA GENERATION DETAILS FOR MSMARCO AND HOTPOTQA EXPERIMENTS IN SECTION 4.2

We use "gpt-4-turbo" to generate paraphrases and contradictions for our experiment in Section 4.2. The prompts we use are in Table 9. We set $temperature = 1$ and $n = 3$ (to generate 3 outputs). Please see Table 8 for some examples of generated paraphrases and contradictions.

## C    ADDITIONAL RELATED WORK

**Complex retrieval tasks**    Information retrieval is a well-studied area Singhal et al. (2001) and there have been many benchmarks for testing retrieval performance such as BEIR Thakur et al. (2021), MTEB Muennighoff et al. (2023), and MIRACL Zhang et al. (2023a). However, most of the datasets, through varying in some degrees, focus only on "retrieving the most similar document". People have noted that there exist some more complex retrieval tasks (e.g. Arguana Wachsmuth et al. (2018) retrieves counter-arguments that refute a query argument), and build retrieval benchmark focusing on complex retrival goals, e.g. BIRCO Wang et al. (2024) and BERRI Asai et al. (2023).

To retrieve according to different instructions, Asai et al. (2023) trains TART, a multi-task retrieval system with task instructions attached as prompts in front of the query content. However, when answering queries, they are still searching for the most similar sentence embedding, though the prompt is different for different tasks. As far as we know, our paper studies the first non-similarity-based search problem.

**Data inconsistency and misinformation detection**    Data inconsistency, refers to the factually incorrectness in the content, might come from different sources, including their natural existence in the corpus Shahi & Nandini (2020); Cui & Lee (2020), data augmentations Jha et al. (2020); Zhou et al. (2022), and pseudo labeling Xie et al. (2020); Wang et al. (2022), which might lead to negative influence if serving as training dataset. There have been a few datasets on detecting the

| Datasets | Orginal | Paraphrase | Contradiction |
|---|---|---|---|
| MSMARCO | In addition to the **high financial value** of higher education, higher education also makes individuals much **more intelligent** than what they would be with just a high school education... | Beyond its **significant monetary worth**, higher education substantially **enhances a person's intelligence** compared to merely completing high school... | Besides the **low financial significance** of higher education, higher education often renders individuals **no more intelligent** than they would be with just a high school education... |
| HotpotQA | Ice hockey is a **contact** team sport played on **ice**, usually in a **rink**, in which two teams of skaters use their **sticks** to shoot a **vulcanized rubber puck** into their opponent's net to score points... | Ice hockey is a **contact** sport where two teams compete on an **ice surface**, typically in a **rink**, using **sticks** to hit a **vulcanized rubber puck** into the opposing team's net to earn points... | Ice hockey is a **non-contact** team sport played on **grass**, often in an **open field**, where two teams of players use their **feet** to kick a **soft leather ball** into their opponent's goal to score points... |

Table 8: Examples of passages from MSMARCO and HotpotQA datasets, with their generated paraphrases, and generated contradictions. Highlighted key-words represent exact matchings or contradictions

| Task | Prompt |
|---|---|
| Generating paraphrases | Paraphrase the given paragraph keeping its original meaning. Do not add information that is not present in the original paragraph. Your response should be as indistinguishable to the original paragraph as possible in terms of length, language style, and format. Begin your answer directly without any introductory words. |
| Generating contradictions | Rewrite the given paragraph to contradict the original content. Ensure the revised paragraph changes the factuality of the original. Your response should be as indistinguishable to the original paragraph as possible in terms of length, language style, and format. Begin your answer directly without any introductory words. |

Table 9: Prompts used to generate paraphrases and contradictions for MSMARCO and HotpotQA documents.

factually wrong information. For example, Laban et al. (2022) detects whether a given summary is consistent with the input document, Shahi & Nandini (2020); Cui & Lee (2020) detects whether a given COVID-19 related news is true or false. Most of these datasets lie in a specific domain and require external knowledge to judge the correctness of each piece of data. On the contrary, the "data inconsistency" notion we consider in our paper doesn't depend on any external knowledge, but is a relationship between different pieces of data in the same corpus. The goal of our method is to find such "contradiction pairs" in corpus efficiently, but not to judge which one is consistent with the real world knowledge.

## D TWO EXAMPLES DEMONSTRATING THE "NON-TRANSITIVITY" OF HOYER SPARSITY AND THE "TRANSITIVITY" OF COSINE FUNCTION

Here, we provide a simple example to demonstrate that using Hoyer sparsity to measure "contradiction" can bypass the challenging scenario for similarity metrics where "A contradicts C, B contradicts C, but A doesn't contradict B". Specifically, Hoyer sparsity satisfies the following "non-transitivity" property.

**Proposition D.1** ("non-transitivity" of hoyer sparsity). *There exist three vectors A, B, and C of dimensionality $d$, satisfying $1 \leq \|A\|_2, \|B\|_2, \|C\|_2 \leq 1 + O(\frac{1}{\sqrt{d}})$, such that $Hoyer(A, C) > 1 - O\left(\frac{1}{\sqrt{d}}\right)$, $Hoyer(B, C) > 1 - O\left(\frac{1}{\sqrt{d}}\right)$, and $Hoyer(A, B) < O(\frac{1}{\sqrt{d}})$*

*Proof.* We construct the following $d$ dimensional vectors where $\epsilon < \frac{1}{d}$ can be any parameter.

$$
\begin{array}{rclllll}
A & = & (1, & 0, & 0, & \ldots, & 0) \\
B & = & (1, & 0, & \epsilon, & \ldots, & \epsilon) \\
C & = & (0, & 1, & 0, & \ldots, & 0)
\end{array}
$$

Then, we calculate their $l_1$ over $l_2$ ratios:

$$\frac{\|A - B\|_1}{\|A - B\|_2} = \sqrt{d - 2}$$

$$\frac{\|A - C\|_1}{\|A - C\|_2} = \sqrt{2}$$

$$\frac{\|B - C\|_1}{\|B - C\|_2} = \frac{2 + (d-2)\epsilon}{\sqrt{2 + (d-2)\epsilon^2}} < \frac{3}{\sqrt{2}}$$

Applying their $l_1$ over $l_2$ ratio bounds to the Hoyer sparsity formula will give us the desired relationship.

$\square$

Next, we provide another example to demonstrate that the cosine function exhibits the following "transitivity" property, which makes it hard to characterize the scenario where "A contradicts C, B contradicts C, but A doesn't contradict B".

**Proposition D.2** ("transitivity" property of cosine function). *Given three unit vectors A, B, and C, if $cos(A, C) \geq 1 - O(\epsilon)$ and $cos(B, C) \geq 1 - O(\epsilon)$, we have $cos(A, B) \geq 1 - O(\epsilon)$*

*Proof.* For any two vectors $X$ and $Y$ with unit norm, we have $cos(X, Y) = 1 - \frac{\|X - Y\|_2^2}{2}$. Because $cos(A, C) \geq 1 - O(\epsilon)$, we have $\|A - C\|_2 \leq O(\sqrt{\epsilon})$. Finally, $cos(A, B) = 1 - \frac{\|A - B\|_2^2}{2} \geq 1 - \frac{(\|A - C\|_2 + \|C - B\|_2)^2}{2} \geq 1 - O(\epsilon)$ $\square$

# E EXPERIMENT COMPARISON WITH METHOD FROM SHI ET AL. (2023)

Shi et al. (2023) proposes "Bipolar-encoder" method to retrieve contradictions from the corpus. They also tested their method on the Arguana dataset but used a different metric, Recall@1. For completeness, we have translated our results into their Recall@1 metric for a fair comparison. As shown in Table 10, both our CL (baseline method) and CL+SparseCL (our method) demonstrate significant improvement over the previous results in Shi et al. (2023).

| Model | Method | Arguana(Recall@1) |
|---|---|---|
| GTE | CL+SparseCL (ours) | 0.629 |
| GTE | CL (baseline) | 0.563 |
| Shi et al. (2023) | Bipolar-encoder | 0.490 |

Table 10: Comparison of experimental results on the Arguana dataset

# F  A CASE STUDY FOR COUNTER-ARGUMENT RETRIEVAL FROM ARGUANA DATASET

In this section we provide an example to illustrate how our sparsity-based retrieval method is better at retrieving counter-arguments. In the setting of the augmented Arguana dataset (see our ablation study in Section 4.5), we selected an example query with an ID "aeghh-pro03a", for which we list the top 10 retrieved passages using the standard cosine similarity score and our sparsity-based score ($\alpha = 1.78$ selected from the dev set). The first five letters of a passage ID represent the argument topic ID; "pro/con" denotes the argument stance; suffix "a/b" indicates the argument and its corresponding counter-argument; "para0/para1/para2" are three paraphrases generated by GPT4.

As shown in Table 11, for the example query "aeghh-pro03a", its correct counter-argument, "aeghh-pro03b" (in red), ranks fourth using the cosine score but first using the cosine + hoyer score. Meanwhile, its paraphrases "aeghh-pro03a-para0/1/2" (in blue) achieve high cosine scores but low sparsity scores.

| Method | | | CL(Cosine)+SparseCL(Hoyer) | | | |
|---|---|---|---|---|---|---|
| Rank | Cosine | Passage ID | Overall | Cosine | Hoyer | Passage ID |
| 1 | 0.940 | **aeghh-pro03a-para0** | 1.683 | 0.794 | 0.499 | **aeghh-pro03b** |
| 2 | 0.926 | **aeghh-pro03a-para2** | 1.644 | 0.719 | 0.519 | aeghh-con02a-para0 |
| 3 | 0.916 | **aeghh-pro03a-para1** | 1.617 | 0.716 | 0.506 | aeghh-con02a-para2 |
| 4 | 0.794 | **aeghh-pro03b** | 1.606 | 0.940 | 0.374 | **aeghh-pro03a-para0** |
| 5 | 0.719 | aeghh-con02a-para0 | 1.602 | 0.718 | 0.496 | aeghh-con02a-para1 |
| 6 | 0.718 | aeghh-con02b | 1.528 | 0.718 | 0.454 | aeghh-con02b |
| 7 | 0.718 | aeghh-con02a-para1 | 1.494 | 0.916 | 0.324 | **aeghh-pro03a-para1** |
| 8 | 0.716 | aeghh-con02a-para2 | 1.426 | 0.926 | 0.280 | **aeghh-pro03a-para2** |
| 9 | 0.696 | aeghh-con02a | 1.396 | 0.669 | 0.408 | dhwif-pro02b |
| 10 | 0.692 | aeghh-pro04a-para0 | 1.344 | 0.628 | 0.402 | thggl-con03b |

Table 11: An example query analysis for counter-argument retrieval. The passage ID in red represents the ground-truth counter-argument, while the passage IDs in blue are paraphrases of the query argument.

# G  HYPER-PARAMETERS FOR TRAINING AND INFERENCE

Here we present the training details (Table 12) for our experiments on Arguana and synthetic HotpotQA and MSMARCO. We report the $\alpha$ parameters tuned on the validation set in Table 13. We search the $\alpha$ parameters from the range $[0, 10]$ by first dividing the range into 10 intervals, calculating the NDCG@10 score on the validation set for each interval's midpoint, and then diving into that interval for a finer search. We stop when the interval range is smaller than $0.01$

| Models | Model Size | Backbone | CL | | SPARSECL | | temp | bz |
|---|---|---|---|---|---|---|---|---|
| | | | ep | lr | ep | lr | | |
| GTE-large-en-v1.5 | 434M | BERT + RoPE + GLU | 1 | 1e-5 | 3 | 2e-5 | 0.01 | 64 |
| UAE-Large-V1 | 335M | BERT | 1 | 2e-5 | 3 | 2e-5 | 0.02 | 64 |
| bge-base-en-v1.5 | 109M | BERT | 1 | 2e-5 | 3 | 2e-5 | 0.02 | 64 |

Table 12: Training parameters for Arguana. We set max sequence length to be 512 for Arguana dataset and 256 for HotpotQA and MSMARCO datasets.

# H  EFFICIENCY TEST OF CROSS-ENCODER AND VECTOR CALCULATION

To further compare the efficiency of cross-encoders and Hoyer sparsity calculations, we perform the following experiments:

| Models | Methods | Arguana | MSMARCO | HotpotQA |
|--------|---------|---------|---------|----------|
| GTE | Zeroshot (Cosine) + SPARSECL(Hoyer) | 0.88 | 2.65 | 2.36 |
|     | CL (Cosine) + SPARSECL(Hoyer) | 0.20 | 0.35 | 5.44 |
| UAE | Zeroshot (Cosine) + SPARSECL(Hoyer) | 0.20 | 1.00 | 1.06 |
|     | CL (Cosine) + SPARSECL(Hoyer) | 0.31 | 1.01 | 1.22 |
| BGE | Zeroshot (Cosine) + SPARSECL(Hoyer) | 0.18 | 2.19 | 4.82 |
|     | CL (Cosine) + SPARSECL(Hoyer) | 0.12 | 2.53 | 3.72 |

Table 13: $\alpha$ choices for different methods and datasets

- We choose "bge-reranker-base" and "bge-reranker-large" to be our cross-encoders. We use them to calculate the similarity between one query from Arguana's test set and 100 documents from Arguana's corpus. We report the average running time of this method for 100 queries.

- We choose "bge-base-en-v1.5" and "bge-large-en-v1.5" to be our bi-encoders. Suppose we have preprocessed all the sentence embeddings. We use it to calculate the Hoyer sparsity between one query embedding from Arguana's test set and 100 document embeddings from Arguana's corpus. We report the average running time of this method for 100 queries.

Please see Table 14 for the running time of different methods. We can see that the calculation of Hoyer sparsity is at least 200 times faster than running a cross-encoder.

| Cross-encoder | Model size | Time |
|---------------|-----------|------|
| bge-reranker-base | 278M | 0.8832s |
| bge-reranker-large | 560M | 1.6022s |
| Bi-encoder | Embedding dimension | Time |
| bge-base-en-v1.5 | 768 | 0.0029s |
| bge-large-en-v1.5 | 1024 | 0.0036s |

Table 14: Average running time for calculating the score functions between one Arguana query and 100 Arguana documents

