# OpenReview forum: "Contradiction Retrieval Via Sparse-Aware Sentence Embedding"
_ICLR.cc/2025/Conference — Submitted to ICLR 2025_

### Official Review · Reviewer_wWVg · 2024-10-24

**Soundness:** 3
**Presentation:** 3
**Contribution:** 3
**Rating:** 6
**Confidence:** 5

**Summary:**

This paper proposes a new task for text retrieval - contradiction retrieval, which aims to retrieve texts contradicting the input one. The author discusses the existing paradigm and claims the intransitive nature of contradiction to be an obstacle to previous contrastive learning scenarios. Following this observation, the authors propose the Hoyer sparse function as the metric for contradiction, which shows some improvements when being merged with semantic similarity scores.

**Strengths:**

1. This work points out a potential issue in learning a specific kind of embedding relationship - contradiction, which might not be well addressed by the existing contrastive learning paradigm.

2. The authors propose an interesting idea for learning sparse representations for contradiction retrieval with the Hoyer sparse function as an intransitive metric.

Overall, the issue is justified with a reasonable method being proposed. This paper includes an interesting idea to make it publish.

**Weaknesses:**

While this paper presents an interesting idea, several major components still exist to make readers agree with the proposed claims.

### There are unaddressed potential opposed claims
1. Instead of learning the contradiction as a metric, it is more natural to reverse the semantics of the context to adapt the contradiction to the similarity. One way to do this is to append instructions to the input (Instructor [1]), such as "Contradicted to: [INPUT]". There can be a baseline (not the original Instructor) learning cosine similarity as the metric follows the traditional contrastive learning paradigm, while the inputs are with the mentioned prefix. Outperforming this baseline can further establish the necessity of learning a sparse metric, otherwise, people may favor context switching more because it can be merged with general similarity learning.

[1] One Embedder, Any Task: Instruction-Finetuned Text Embeddings @ ACL2023 Findings

2. Is there a stable selection of \alpha? While direct semantic similarity plays an important role in contradiction retrieval in the benchmarks mentioned in the paper. The weight on the semantic similarity is quite unstable from my viewpoint - contradiction shall be not so relevant with the semantic similarity, for instance

Case1:
Text1: "Dinosaurs live on earth today." Text2: "Dinosaurs don't live on earth today."

Case2:
Text1: "Dinosaurs live on earth today." Text2: "A huge asteroid hit the earth 65 million years ago and caused the extinction of dinosaurs."

While the cases above are somehow extreme, this still indicates how contradiction is independent of the direct textual similarity. From some readers' view, the semantic similarity should be treated as a threshold (not totally irrelevant) rather than a weighted term. So,

- Why do you think semantic similarity should still play an important role in contradiction retrieval? I think showing some real-world distribution in the corpus can lead to some help.

### Claim support is incomplete
1. An important claim of this paper is the advantage of the Hoyer function over cosine similarity because of its intransitive nature. However, this point is not supported in the content because there lack of baselines trained on the reversed similarity labels. Please consider adding this as a baseline to your main table.

- How would you define the contradiction level with non-statements like questions?

2. The performance improvement on synthesized datasets is much higher than Arguana. Based on the prompts and cases provided in the Appendix, I feel the generated contradictions are somehow favored by the proposed paradigm - mostly similar with some sparse differences. However, as mentioned above, contradiction can happen between texts with very different "length", "style", "format", so I feel this synthesis is oversimplifying the problem and making the dataset favor the proposed method. I would suggest the authors allow the synthesizer to generate text pairs with very different other attributes to further validate the too significant improvement.

- Why don't you include NLI datasets as a source of benchmarking, they directly include contradicted text pairs.

### Not enough content
1. Lack of baselines. The main comparison only includes a contrastively learned encoder, which is even unintended for the proposed contradiction retrieval task. Even though contradiction retrieval is a new task, there should be some simple baselines as mentioned above. You can also consider adding the original Instructor as a baseline, though I feel I will not perform well.

2. Lack of task significance justification. As a newly proposed task focusing on a rather narrow scope, the paper does not fully justify the importance of further digging into this task. One way to further support the importance of contradiction retrieval is to select one of the tasks mentioned in the related works to show they can be improved with contradiction retrieval.

Overall, while this paper proposes a reasonable way to address a potential issue in text embedding learning, I feel this paper has to be significantly polished to reach publication quality. Thus, I am opposed to an immediate acceptance of this paper.

**Questions:**

My questions are corresponded with the weaknesses above.

---

> ### Author Response · Authors · 2024-11-23
>
> We thank Reviewer wWVg for providing valuable feedback!
>
> Append instructions to the input:
>
> Thanks for proposing this baseline. First, adding a task-specific prompt in front of the input task is already used by models on the MTEB leaderboard. For example, SFR-Mistral-Embedding appends 'Given a claim, find documents that refute the claim' in front of the input when testing Arguana. Their reported result is 67.17, while the models we chose, 'uae-large-v1' and 'gte-large-en-v1.5,' have zero-shot scores of 68.3 and 72.5, respectively. This indicates that appending a specific prompt does not improve performance for the counter-argument retrieval task. Readers can refer to the results on MTEB for other models with similar prompts, such as Instructor-xl, which has a score of 55.65.
> Second, using language models to generate a contradiction and then applying similarity search is indeed a baseline method, though it requires significantly higher computational resources to generate the contradiction before performing the search. We have already experimented with this in our ablation study. Please refer to the paragraph 'Different Scoring Function for Contradiction Retrieval' and Figure 5 at the end of page 9.
>
> Stable choice of alpha
>
> Thanks for proposing this interesting question and example!
> First, we want to clarify that contradiction is indeed a somewhat subjective opinion that varies across different minds. As mentioned in your example, what about
> “A huge asteroid hit the earth 65 million years ago and killed many dinosaurs.” Or “Some scientists hypothesize that a huge asteroid hit the earth 65 million years ago and caused the extinction of dinosaurs, but this hypothesis hasn’t been fully verified yet.” We believe that different people make different judgments about whether these are contradictions or not.
> Second, we want to note that we use two different models to provide the cosine similarity score and the sparsity score. Since the two models are trained independently and potentially with different backbones and temperatures, it is not meaningful to directly sum them without proper scaling. For example, when we trained 'bge-base-v1.5' on Arguana with temperatures of 0.02 and 0.05, the average cosine similarity between ground truth pairs was 0.81 and 0.69, respectively. Therefore, cosine similarity scores are unstable given different temperatures, but they are effective at providing relative ranking for retrieval tasks. This motivates us to use a weighted average to combine the cosine similarity score and sparsity score.
>
> Why do you think semantic similarity should play a role in contradiction retrieval
>
> Please refer to Appendix E for our case study on how the similarity and sparsity scores are used in a typical contradiction retrieval example.

---

> > ### Author Response · Authors · 2024-11-23
> >
> > Lack of baseline trained on reversed similarity labels
> >
> > Please see our answer for your weakness 1. We have provided the baseline of “generating the query’s contradiction and then applying standard similarity search” in our Table 5.
> >
> > How would you define the contradiction level with non-statements like questions
> >
> > That’s a good question. As far as we know, even the standard QA definition is not well-defined in the embedding space. Our research provides the first non-similarity-based definition of contradictions. We leave other fine-grained definitions between passages and queries as future work.
> >
> > Improvement on synthesized datasets is much higher than Arguana:
> >
> > We provide two reasons: First, we inject more paraphrases into the corpus to confuse the similarity search, which causes the standard similarity search to perform poorly in this case. Please refer to our dataset construction and result analysis (lines 359-364) in Section 4.2. Second, our synthetic HotpotQA and MSMARCO datasets contain more balanced paraphrases and contradiction pairs, providing 88k (original sentence, positive example, negative example) tuples for fine-tuning. For Arguana, we were only able to gather 14k such tuples for fine-tuning.
> >
> > Include NLI as a source of benchmarking
> >
> > This is a good question. We have considered both SNLI and MNLI to test our method in the earlier phase, but they are not suitable for different reasons:
> >
> > SNLI: The sentences there are very short and too general, which means that a sentence in one contradiction pair can also be a contradiction in another pair. This property is not suitable for retrieval tasks, because we want the contradiction corpus to be unique and fully specified by the ground truth. For example:
> > Pair A: (xxx, A blond man drinking water from a fountain.)
> > Pair B: (xxx, A man is drinking juice.)
> > In this situation, the second sentence in pair B is a valid contradiction for pair A, but it is not reflected in the ground truth label if we were to transform it into a retrieval task.
> >
> > MNLI: There, every contradiction pair (sent A, sent B) is irrelevant to each other. If we treat sent A as the query, then its contradictory sentence, sent B, is also the most similar sentence in the corpus. Therefore, the dataset won't benefit from contradiction retrieval compared to directly searching for the most similar sentence. However, in the Arguana dataset, many arguments from the same debate exist in the corpus.
> >
> > Lack of baselines
> >
> > We respectfully thank you for the comment.
> > First, adding task-specific prompts like 'Given a claim, find documents that refute the claim' is already employed by models on the MTEB leaderboard, such as SFR-Mistral-Embedding, and similar prompts are used by Instructor. Please refer to the MTEB leaderboard for their performance on the Arguana dataset.
> > Second, we have provided the baseline of 'generating the query’s contradiction and then applying standard similarity search' in Table 5.
> > Third, we want to note that the paper [Hongguang Shi, Shuirong Cao, and Cam-Tu Nguyen. 'Revisiting the Role of Similarity and Dissimilarity in Best Counter-Argument Retrieval'] introduced a special method for contradiction retrieval, which we have compared in the appendix. For the sake of uniformity, we didn’t include it in the main table because they use a different metric from the one reported on the MTEB.
> > Lastly, we want to emphasize that the lack of a baseline for the contradiction retrieval task is a motivation for our work, as it justifies the need for further research.
> >
> > Lack of task significance justification
> >
> > Please see our application in Section 4.4, where we apply our method to find inconsistencies within a corpus. Our results show that SPARSECL can clean up inconsistencies within the corpus and recover QA retrieval accuracy (see Table 3).

---

> > > ### Author Response · Authors · 2024-11-24
> > >
> > > Dear reviewer,
> > >
> > > Thank you for your insightful feedback on our work! We've carefully worked to address your comments/questions. Are there any further questions or concerns we should discuss?

---

> > > > ### Comment · Reviewer_wWVg · 2024-11-24
> > > >
> > > > For NLI datasets, you do not need to follow the retrieval task to benchmark them but can calculate the score inside a batch, for instance in SNLI:
> > > >
> > > > **premise:** A person on a horse jumps over a broken down airplane.
> > > >
> > > > **hypothesis 1:** A person is training his horse for a competition.
> > > >
> > > > **hypothesis 2:** A person is at a diner, ordering an omelette.
> > > >
> > > > **hypothesis 3:** A person is outdoors, on a horse.
> > > >
> > > > Then you can check whether hypothesis 2 is assigned the highest contradiction score.

---

> > > > > ### Author Response · Authors · 2024-11-25
> > > > > **Experiments on SNLI**
> > > > >
> > > > > Thank you for your suggestion. According to your proposal, we have performed an experiment on the SNLI dataset to prove the validity of our method there.
> > > > >
> > > > > As suggested, we only extract those entailment and contradiction pairs from the SNLI dataset. We finetune our BGE model using standard contrastive learning and our SparseCL method.
> > > > >
> > > > > Similar to what we did in our main experiment, we construct a training tuple like the following:
> > > > > 	(Original text, Contradiction (positive example), Entailment (negative example))
> > > > >
> > > > > We first measure the average cosine similarity among entailment, contradiction, and random pairs before training.
> > > > > ||contradiction|entailment|random|
> > > > > |-|-|-|-|
> > > > > |avg cosine similarity|0.546|0.769|0.376|
> > > > >
> > > > >
> > > > > After performing contrastive learning, we can get the following statistics:
> > > > > ||contradiction|entailment|random|
> > > > > |-|-|-|-|
> > > > > |avg cosine similarity|0.885|0.886|0.777|
> > > > >
> > > > >
> > > > > After performing SparseCL, we can get the following statistics:
> > > > > ||contradiction|entailment|random|
> > > > > |-|-|-|-|
> > > > > |avg hoyer sparsity|0.314|0.278|0.231|
> > > > >
> > > > >
> > > > > By applying a threshold on hoyer sparsity to do entailment/contradiction classification, we can get accuracy 0.647, where the threshold is selected on the validation set. However, the standard contrastive learning doesn’t work here, whose binary classification accuracy is 0.507. That’s why we don’t think standard contrastive learning makes sense for contradiction retrieval task, but we still perform it as requested.
> > > > >
> > > > > We admit that for the simple entailment/contradiction classification task, relying only on cosine similarity is good enough, as entailment are always closer to the original sentence than contradictions, while the point here is to show that our method based on hoyer sparsity also works for NLI datasets.
> > > > >
> > > > > Additionally, we want to point out that relying on cosine similarity to identify contradictions from a large corpus is hard because we can observe that the cosine similarities within contradictions are between entailments and random or unrelated pairs. However, by applying SparseCL, we manage to let contradiction pairs stand out with the highest hoyer sparsity score. Our results here are consistent with what we reported in our paper.
> > > > >
> > > > > Thanks again for your insightful feedback on our work! We've carefully worked to address your comments/questions. Are there any further questions or concerns we should discuss?

---

> > > > ### Comment · Reviewer_wWVg · 2024-11-24
> > > >
> > > > For the instruction appending baseline, I am still not convinced by your reasoning for not including them in the main table - the baseline you mentioned has not been trained on contradiction instructions. However, this type of supervision is applied in your embedding fine-tuning, which makes the comparison unfair.

---

> ### Comment · Reviewer_wWVg · 2024-11-25
>
> Thanks for your new results. I think the result will support the conclusion of this paper with more references to validated resources. However, there is a critical concern that restrains me from accepting this paper - the necessity of contradiction retrieval itself as a task requiring special methods. As mentioned in our previous discussion, contradiction retrieval can be formalized as regular retrieval with context modification (such as appending instructions). It is not sure whether contraction retrieval can be unified with regular retrieval in fine-tuning. I appreciate the efforts the authors devote to the discussion period, I will directly raise my rating to **accept** once the two following modifications have been done:
>
> - Running an experiment with fine-tuning text embeddings (such as BGE) on contradiction pairs formalized as regular retrieval pairs. e.g. Contradict(Statement1, Statement2) -> Similar("Not true: {Statement 1}", "{Statement 2}"). Then test the retriever with "Not true: {Statement}".
>
> - Add the SNLI benchmark result to Table 1. At least one other pair-matching benchmark should be considered. I suggest the Winograd (https://huggingface.co/datasets/ErnestSDavis/winograd_wsc) challenge, which includes challenging NLI for contradiction.

---

> > ### Author Response · Authors · 2024-11-27
> >
> > Thanks for your effort in discussing with us. We have provided the two experiments in our PDF according to your suggestions. The new added contents are in red text. **Please refer to our PDF for full results and analysis**
> >
> > 1. We have fine-tuned BGE by appending the “Not true: ” prompt in the front of the query. Please see our results and analysis in our Table 5.
> >
> > |Model|Method|Arguana|
> > |---|---|---|
> > |BGE|Prompt + Zeroshot(Cosine)|0.657|
> > |BGE|Zeroshot (Cosine)|0.658|
> > |BGE|Prompt + CL(Cosine)|0.645|
> > |BGE|CL (Cosine)|0.687|
> >
> > Here “Prompt” means appending the “Not true: ” prompt in the front of the query text. “Zeroshot” refers to direct testing and “CL” refer to standard contrastive learning.
> > From the table, we observe that “Prompt” doesn’t improve much upon standard similarity search. For the “Prompt + CL (Cosine)” method, fine-tuning with the appended prompt even results in a performance drop. During the training process, we observed overfitting and hypothesize that the special prompt “Not true:” introduces a shortcut, making it easier for the model to learn whether a text belongs to the “argument” class or the “counter-argument” class. However, this class information is not useful when identifying pairwise contradiction relationships.
> >
> >
> > 2. Thank you for your suggestion in applying our SparseCL method to NLI datasets. We have performed experiments on SNLI and MNLI datasets (Sorry, we didn’t use winograd because their dataset doesn’t provide “entailment” pairs for fine-tuning or testing. As far as we can see, winograd provides a sentence with two opposite options, which only generates a pair of contradictions, but no entailment can be derived from that dataset.) Please see the results in Table 7 of Appendix A.
> >
> > As an application of our SPARSECL method, we demonstrate that our method can be useful for distinguishing entailments and contradictions in natural language inference datasets. For SNLI Bowman et al. (2015) and MNLI Williams et al. (2018) datasets, we extract entailment and contradiction pairs, fine-tune using standard contrastive learning and our SPARSECL, and then report the average cosine similarity / Hoyer sparsity score between entailments, contradictions, and random pairs.
> >
> >
> > |---|---|contradiction|entailment|random|
> > |---|---|---|---|---|
> > |SNLI|Zeroshot (Cosine)|0.546|0.769|0.376|
> > |SNLI|CL (Cosine)|0.885|0.886|0.777|
> > |SNLI|SparseCL(Hoyer)|0.376|0.347|0.228|
> > |MNLI|Zeroshot (Cosine)|0.659|0.818|0.378|
> > |MNLI|CL (Cosine)|0.919|0.917|0.733|
> > |MNLI|SparseCL (Hoyer)|0.422|0.364|0.244|
> >
> >
> > We can observe from the table that, in the zeroshot setting, the average cosine similarity of contradiction pairs lies between the ranges of random and entailment pairs. For the fine-tuned model using standard contrastive learning (CL), the average cosine similarity of contradiction pairs is almost indistinguishable from that of entailment pairs. Finally, after being fine-tuned using SPARSECL, the model exhibits higher average Hoyer sparsity scores for contradiction pairs compared to other two types of relationships.
> >
> > We are happy to discuss further if reviewer wWVg has any concern.

---

> > > ### Comment · Reviewer_wWVg · 2024-11-27
> > >
> > > Thanks for the rebuttal, my major concerns are well-addressed:
> > >
> > > - More curated benchmarks are involved.
> > >
> > > - Important baselines are added to the main experiments, which shows the task cannot be simply unified by instruction-following.
> > >
> > > - Other discussions have been made, such as alpha selection.
> > >
> > > Now I vote to **accept** this paper based on the significant changes the authors have made. Besides I think the paper will further benefit from mentioning how contradiction retrieval cannot be formalized as traditional retrieval.

---

### Official Review · Reviewer_83EJ · 2024-11-03

**Soundness:** 2
**Presentation:** 3
**Contribution:** 2
**Rating:** 5
**Confidence:** 4

**Summary:**

This paper works on contradiction retrieval, which is to identify and extract documents that disagree or refute the content of a query. Existing methods fail to capture the essence of contradiction due to its similarity comparison nature, or has efficiency issues. The paper proposes SparseCL to address these challenges, which leverages sentence embeddings designed to preserve subtle, contradictory nuances between sentences. This method could identify contradictory documents with simple vector calculations. The proposed method was verified on the Arguana dataset and synthetic contradictions from MSMARCO and HotPotQA. It has another application, which is cleaning corrupted corpora to restore high-quality QA retrieval.

**Strengths:**

- The experiments show that the proposed method (SparseCL) is able to maintain high performance on tested datasets compared with crossencoders while it has less computational requirement by utilizing a simple vector calculation.
- The idea of using sparsity metric to measure contradiction plus normal sentence embedding similarity is novel. As the paper mentions in line 347, there were no method for retrieving contradictions that only rely on sentence embeddings.

**Weaknesses:**

- In line 229-232: The scalar $\alpha$ in scoring function F(p1, p2) requires tuning on target dataset, which limits the generalization of the method. It shows the proposed method can’t adapt to unseen data (domain shift) automatically.
- The scale of the Arguana dataset is small, with less than 7k argument-counterargument pairs.
- To address the generalization of embeddings obtained from model fine-tuning, the paper reports results from training on MS MARCO and testing on HotpotQA, and vice versa. HotpotQA consists of corpora sourced from Wikipedia, whereas MS MARCO is derived from Bing search logs, which likely include some Wikipedia documents. Therefore, the paper should first address potential data contamination to ensure that the model is tested on truly unseen data before claiming generalization of the embeddings.

**Questions:**

- Since SparseCL could do contradiction retrieval, it’s also possible for it to be used as a contradiction detection model (or a contradiction scoring model). For example, there are many datasets in NLI and Fact Checking domains. It would be interesting to see if this method works on these datasets.
- In line 237-238: Please check reference format.

---

> ### Author Response · Authors · 2024-11-23
>
> We thank Reviewer 83EJ for providing valuable feedback!
>
> The scalar alpha requires tuning on target dataset:
>
> Thanks for proposing this interesting question!
> First, we want to state that contradiction is indeed a somewhat subjective opinion that varies across different individuals. As mentioned by another reviewer, consider the query text, 'Dinosaurs live on Earth today,' and two other texts: 'A huge asteroid hit the Earth 65 million years ago and killed many dinosaurs.' and 'Some scientists hypothesize that a huge asteroid hit the Earth 65 million years ago and caused the extinction of dinosaurs, but this hypothesis hasn’t been fully verified yet.' These texts may seem to contradict the query, depending on individual perspectives. We believe that different people make different judgments about whether these are contradictions.
>
> Second: We want to note that we use two different models to provide the cosine similarity score and the sparsity score. Since the two models are trained independently and potentially with different backbones and temperatures, it is not meaningful to directly sum them up without proper scaling. For example, when we trained 'bge-base-v1.5' on Arguana with temperatures of 0.02 and 0.05, the average cosine similarity between ground truth pairs was 0.81 and 0.69, respectively. Therefore, cosine similarity scores are unstable given different temperatures, but they are effective at providing relative ranking for retrieval tasks. This motivates us to use a weighted average to combine the cosine similarity score and sparsity score.
>
> The Arguana dataset is small: real-world dataset
>
> Arguana (Retrieval of the Best Counterargument without Prior Topic Knowledge, ACL 2018) is a widely used argument/counter-argument dataset with over 100 citations. As far as we know, Arguana is the only public dataset suitable for our test (both positive and negative documents are included in the corpus). For this reason, we created synthetic datasets to perform additional tests. Finally, we would like to note that many popular retrieval datasets are on a small scale. For example, among the popular BEIR benchmark datasets, NFCorpus (corpus size: 3.6K) and SciFact (corpus size: 5K).
>
> Potential data contamination:
>
> Thank you for raising this concern. We have verified that there is no overlap between the MSMARCO and HotpotQA datasets we have created. Therefore, we believe the zero-shot experiment we conducted is sound. In the following, we provide the number of unique passages for MSMARCO and HotpotQA:
> | Dataset | Train | dev | test|
> |-------|-------|-------|------|
> |MSMARCO | 59500 | 5950 | 44101|
> |HotpotQA | 59074 | 5896 | 81673|
>
> Contradiction detection model:
>
> Thanks for the comment. We agree that our scoring function can be used to classify whether a pair of passages contradicts each other. For example, we evaluate the classification accuracy on the Arguana dataset, where we trained our model. We select half of the test data to be true argument/counter-argument pairs and the other half to be arguments paired with a random argument from its top 10 most similar arguments, ensuring that the labels are balanced. The accuracy is listed below:
> | Method | Acc |
> | ---------- | ----- |
> | CL (cosine) | 0.661|
> | CL (cosine) + SparseCL (Hoyer) | 0.697 |
> For our weighted score function we directly copy the alpha choice from Table 12.
>
> Thanks for pointing out the format issue. We have revised that reference format in the uploaded version.

---

> > ### Author Response · Authors · 2024-11-24
> >
> > Dear reviewer,
> >
> > Thank you for your insightful feedback on our work! We've carefully worked to address your comments/questions. Are there any further questions or concerns we should discuss?

---

> > ### Comment · Reviewer_83EJ · 2024-11-25
> >
> > - Thanks for the response from the authors!
> > - Scalar $\alpha$
> >     - I believe those topics have been thoroughly discussed in the Natural Language Inference (NLI) tasks. For example, “hypothesis hasn’t been fully verified yet” corresponds to the label “neutral” in NLI. The explanation from the authors is not convincing. There could be variance due to the cognition difference by human annotators. But that’s not the reason for tuning $\alpha$ for each dataset. Such requirement limits the generalization of the method and its application (on unseen data). Authors may want to show the result without tuning $\alpha$ for individual dataset and evaluate it on all the datasets mentioned in the paper.
> > - Datasets
> >     - The authors address some concerns regarding the datasets. But I’m still worried about the generalization of this work. More datasets such as those from NLI may be required.
> > - I will update the scores accordingly

---

> ### Author Response · Authors · 2024-11-25
> **Experiments on SNLI**
>
> Thank you for your suggestion. According to your proposal, we have performed an experiment on the SNLI dataset to prove the validity of our method there.
>
> As suggested by reviewer wWVg, we only extract those entailment and contradiction pairs from the SNLI dataset. We finetune our BGE model using standard contrastive learning and our SparseCL method.
>
> Similar to what we did in our main experiment, we construct a training tuple like the following:
> 	(Original text, Contradiction (positive example), Entailment (negative example))
>
> We first measure the average cosine similarity among entailment, contradiction, and random pairs before training.
> ||contradiction|entailment|random|
> |-|-|-|-|
> |avg cosine similarity|0.546|0.769|0.376|
>
>
> After performing contrastive learning, we can get the following statistics:
> ||contradiction|entailment|random|
> |-|-|-|-|
> |avg cosine similarity|0.885|0.886|0.777|
>
>
> After performing SparseCL, we can get the following statistics:
> ||contradiction|entailment|random|
> |-|-|-|-|
> |avg hoyer sparsity|0.314|0.278|0.231|
>
>
> By applying a threshold on hoyer sparsity to do entailment/contradiction classification, we can get accuracy 0.647, where the threshold is selected on the validation set. However, the standard contrastive learning doesn’t work here, whose binary classification accuracy is 0.507. That’s why we don’t think standard contrastive learning makes sense for contradiction retrieval task, but we still perform it as requested.
>
> We admit that for the simple entailment/contradiction classification task, relying only on cosine similarity is good enough, as entailment are always closer to the original sentence than contradictions, while the point here is to show that our method based on hoyer sparsity also works for NLI datasets.
>
> Additionally, we want to point out that relying on cosine similarity to identify contradictions from a large corpus is hard because we can observe that the cosine similarities within contradictions are between entailments and random or unrelated pairs. However, by applying SparseCL, we manage to let contradiction pairs stand out with the highest hoyer sparsity score. Our results here are consistent with what we reported in our paper.
>
> Thanks again for your insightful feedback on our work! We've carefully worked to address your comments/questions. Are there any further questions or concerns we should discuss?

---

> > ### Author Response · Authors · 2024-11-27
> > **More experiments on NLI dataset**
> >
> > Thank you for your suggestion in applying our SparseCL method to NLI datasets. We have performed more experiments on SNLI and MNLI datasets. Please see the updated results in Table 7 of Appendix A.
> >
> > As an application of our SPARSECL method, we demonstrate that our method can be useful for distinguishing entailments and contradictions in natural language inference datasets. For SNLI Bowman et al. (2015) and MNLI Williams et al. (2018) datasets, we extract entailment and contradiction pairs, fine-tune using standard contrastive learning and our SPARSECL, and then report the average cosine similarity / Hoyer sparsity score between entailments, contradictions, and random pairs.
> >
> >
> > |---|---|contradiction|entailment|random|
> > |---|---|---|---|---|
> > |SNLI|Zeroshot (Cosine)|0.546|0.769|0.376|
> > |SNLI|CL (Cosine)|0.885|0.886|0.777|
> > |SNLI|SparseCL(Hoyer)|0.376|0.347|0.228|
> > |MNLI|Zeroshot (Cosine)|0.659|0.818|0.378|
> > |MNLI|CL (Cosine)|0.919|0.917|0.733|
> > |MNLI|SparseCL (Hoyer)|0.422|0.364|0.244|
> >
> >
> > We can observe from the table that, in the zeroshot setting, the average cosine similarity of contradiction pairs lies between the ranges of random and entailment pairs. For the fine-tuned model using standard contrastive learning (CL), the average cosine similarity of contradiction pairs is almost indistinguishable from that of entailment pairs. Finally, after being fine-tuned using SPARSECL, the model exhibits higher average Hoyer sparsity scores for contradiction pairs compared to other two types of relationships.

---

> > > ### Author Response · Authors · 2024-11-27
> > >
> > > Thank you for your suggestion. We have performed the following zeroshot generalization experiment to show that our SparseCL method is capable of generalizing to unseen dataset without tuning alpha.
> > >
> > > Specifically, following the zero shot generalization test we perform in Table 2, where we train SparseCL on MSMARCO/HotpotQA dataset and then test on the other dataset. We conduct the test with the $\alpha$ tailored to the original training dataset (2.19 for model trained on MSMARCO and 4.82 for model trained on HotpotQA). From the table below, we can observe that the performance is stable even if we don’t tune $\alpha$ for each dataset.
> > >
> > > This experiment indicates the generalizability of our proposed method. Thanks the reviewer again for pointing out this issue!
> > >
> > > |Model|Method|Train Dataset|Test Dataset|NDCG@10|
> > > |---|---|---|---|---|
> > > |BGE|Zeroshot (Cosine) + SparseCL (Hoyer) (fixed $\alpha$)|MSMARCO|HotpotQA|0.881|
> > > |BGE|Zeroshot (Cosine) + SparseCL (Hoyer) (fixed $\alpha$)|HotpotQA|MSMARCO|0.822|
> > > |BGE|Zeroshot (Cosine) + SparseCL (Hoyer) (tuned $\alpha$)|MSMARCO|HotpotQA|0.886|
> > > |BGE|Zeroshot (Cosine) + SparseCL (Hoyer) (tuned $\alpha$)|HotpotQA|MSMARCO|0.877|
> > > |BGE|Zeroshot (Cosine) + SparseCL (Hoyer)|HotpotQA|HotpotQA|0.967|
> > > |BGE|Zeroshot (Cosine) + SparseCL (Hoyer)|MSMARCO|MSMARCO|0.909|
> > > |BGE|Zeroshot (Cosine)|N/A|HotpotQA|0.595|
> > > |BGE|Zeroshot (Cosine)|N/A|MSMARCO|0.600|

---

### Official Review · Reviewer_7xxh · 2024-11-03

**Soundness:** 3
**Presentation:** 3
**Contribution:** 3
**Rating:** 6
**Confidence:** 4

**Summary:**

This paper addresses the challenge of contradiction retrieval, which involves identifying documents that explicitly refute or disagree with a given query. Traditional approaches, such as similarity search and cross-encoder models, have significant limitations in this context: similarity-based methods struggle to capture contradictions, while cross-encoders are computationally costly, particularly for large datasets. To overcome these challenges, the authors propose a framework, SPARSECL, that combines specially trained sparse-aware sentence embeddings with a combined metric based on cosine similarity and the Hoyer measure of sparsity. This combination allows for efficient and accurate contradiction retrieval without the computational burden of traditional models. Experiments on three datasets demonstrate the effectiveness of SPARSECL when plugged into different LLM-based retrievers.

**Strengths:**

+ The problem of contradiction retrieval is important and has practical value (e.g., in scientific claim verification). The authors point out the limitations of existing bi-encoder and cross-encoder approaches and propose corresponding techniques to overcome the challenges.

+ The idea of sparse-aware sentence embeddings and the use of the Hoyer measure are intuitive and well-motivated. To be specific, The Hoyer measure is non-transitive, meaning it can effectively distinguish between sentences that are indirectly related but still contradictory.

+ Experiments are comprehensive. The proposed SPARSECL technique is plugged into various LLM-based retrievers (i.e., UAE-Large-V1, bge-base-en-v1.5, and gte-large-en-v1.5) to validate its generalizability. Besides performance comparison, the authors also conduct ablation studies to verify some of their design choices.

**Weaknesses:**

- Statistical significance tests are not conducted. It is not clear whether the gaps between X+Zeroshot and X+Zeroshot+SPARSECL are statistically significant or not (X=UAE-Large-V1, bge-base-en-v1.5, and gte-large-en-v1.5). In fact, the improvement is subtle in some columns in Table 1.

- Only one real dataset (i.e., Arguana) is examined in the experiments. The other two used datasets are synthesized from MSMARCO and HotpotQA, respectively, using GPT-4. It would be valuable to explore other real datasets from other domains, such as scientific claims (e.g., SciFact [1]).

[1] Fact or Fiction: Verifying Scientific Claims. EMNLP 2020.

**Questions:**

- Could you conduct a statistical significance test (e.g., two-tailed t-test) to compare X+Zeroshot with X+Zeroshot+SPARSECL, and report the p-values?

- Could you try SPARSECL on the SciFact benchmark dataset (either its original version or the BEIR version) to retrieve paper abstracts that refute a given query claim?

---

> ### Author Response · Authors · 2024-11-23
>
> We thank Reviewer 7xxh for providing valuable feedback!
>
> Statistical significance tests are not conducted:
>
> We thank the reviewers to point out that the current relative gain on the Arguana dataset might not be significant. So we choose different random seeds to re-run the experiments.
> | Model | Method                                    | Arguana | Values                | Mean  | Std Dev | p-value (vs previous row) |
> |-------|------------------------------------------|---------|-----------------------|-------|---------|---------------------------|
> | BGE   | Zeroshot (Cosine)                        | 0.658   | 0.658, 0.658, 0.658  | 0.658 | 0.000   | -                         |
> | BGE   | Zeroshot (Cosine) + SPARSECL(Hoyer)      | 0.704   | 0.704, 0.713, 0.708  | 0.708 | 0.005   | 0.0027                    |
> | BGE   | CL (Cosine)                              | 0.687   | 0.687, 0.659, 0.673  | 0.673 | 0.014   | -                         |
> | BGE   | CL (Cosine) + SPARSECL(Hoyer)            | 0.722   | 0.722, 0.731, 0.719  | 0.724 | 0.006   | 0.013                     |
>
> We computed the p-value of the difference between our method LINE2 and LINE1, and LINE4 and LINE3, and the p-value is 0.003 and 0.01, respectively, indicating that the relative gain is significant.
>
> Only one real dataset is examined in the experiments:
>
> As far as we know, Arguana is the only direct task where we can verify our method and compare it with results from the literature. We admit that there are some other similar fact verification datasets like FEVER, SciFact, while they are not ideal for testing our method because we need BOTH supporting and contradicting documents to exist in the corpus. If only either supporting or contradicting evidence exists in the corpus, standard similarity retrieval can already extract the related one. Experiment on such datasets requires manually injecting more conflicting documents to the corpus, which is exactly what we have done to HotpotQA and MSMARCO datasets.

---

> ### Author Response · Authors · 2024-12-02
> **More experiments on other real benchmark datasets**
>
> Thank you for your suggestion in applying our SparseCL method to other datasets. We have performed more experiments on SNLI and MNLI datasets. Please see the updated results in Table 7 of Appendix A.
>
> As an application of our SPARSECL method, we demonstrate that our method can be useful for distinguishing entailments and contradictions in natural language inference datasets. For SNLI Bowman et al. (2015) and MNLI Williams et al. (2018) datasets, we extract entailment and contradiction pairs, fine-tune using standard contrastive learning and our SPARSECL, and then report the average cosine similarity / Hoyer sparsity score between entailments, contradictions, and random pairs.
>
>
> |---|---|contradiction|entailment|random|
> |---|---|---|---|---|
> |SNLI|Zeroshot (Cosine)|0.546|0.769|0.376|
> |SNLI|CL (Cosine)|0.885|0.886|0.777|
> |SNLI|SparseCL(Hoyer)|0.376|0.347|0.228|
> |MNLI|Zeroshot (Cosine)|0.659|0.818|0.378|
> |MNLI|CL (Cosine)|0.919|0.917|0.733|
> |MNLI|SparseCL (Hoyer)|0.422|0.364|0.244|
>
>
> We can observe from the table that, in the zeroshot setting, the average cosine similarity of contradiction pairs lies between the ranges of random and entailment pairs. For the fine-tuned model using standard contrastive learning (CL), the average cosine similarity of contradiction pairs is almost indistinguishable from that of entailment pairs. Finally, after being fine-tuned using SPARSECL, the model exhibits higher average Hoyer sparsity scores for contradiction pairs compared to other two types of relationships.

---

### Official Review · Reviewer_MNad · 2024-11-06

**Soundness:** 3
**Presentation:** 2
**Contribution:** 3
**Rating:** 6
**Confidence:** 4

**Summary:**

This work introduces a new loss for training sentence embeddings towards the task of contradiction retrieval. Unlike the conventional SentenceBert setting, two sentences should have high embedding similarity if they contradict with each other, while having low similarity if one is the other's paraphrasing. This work demonstrates that the traditional contrastive learning does not work well for the contradiction retrieval setting, and proposes a new contrastive loss based on Hoyer sparsity distance, which encourages the sparsity of the difference between two contradicting embeddings.

Experiments are conducted on two specific tasks: 1) evaluation on Arguana dataset, a counter-argument retrieval dataset in MTEB benchmark; 2) evaluation on synthetic passages paraphrased from MSMARCO and HotpotQA for contradiction retrieval. Results suggest that the new model is able to bring consistent improvement over the conventional contrastive model, especially for the second task, where finetuning with conventional contrastive is not effective.

**Strengths:**

- A new loss is proposed to train sentence embeddings for contradiction retrieval. The loss is based on embedding sparsity, motivated by the fact that two contradicting sentences may appear similar, thus traditional contrastive may not work as well.

- The proposed method is shown empirically effective in two task settings, surpassing baselines by large margins.

**Weaknesses:**

- Though the motivation is mentioned in the paper, I think there needs more explanation and analysis on why sparsity-based distance works better.

- - The Introduction Section states that traditional cosine-similarity metric is transitive. However, it is a wrong statement, as it is not transitive in vector space, especially for high-dimensional space.

  - The experiments could give more quantitative analysis or qualitative examples on the role that the sparsity loss is playing.

- The paper needs more polishing for formal writing. There are also typos, e.g. Line225 "can can".

**Questions:**

For the synthetic task on MSMARCO and HotpotQA, does each train/dev/test split stem from a different set of original passages, or there can be overlap?

---

> ### Author Response · Authors · 2024-11-23
>
> We thank Reviewer MNad for providing valuable feedback!
>
> Regarding transitivity: Thanks for pointing out this! We admit that cosine similarity is not transitive. We want to comment that for normalized embeddings, ranking by cosine similarity is equivalent to ranking by decreasing l2 distance, where l2 distance is transitive. We will make this claim rigorous in the final version of our paper.
>
> More quantitative analysis or qualitative examples: Thanks for the comment. We actually have some qualitative examples as case studies. Please see appendix E for a case study, where we give an example to show how sparsity score helps retrieving contradicting documents.
>
> Does each train/dev/test split stem from a different set of original passages: Thanks for bringing out this concern. We noticed the overlap issue when creating the dataset and have filtered them out, so there is no overlap between the train/dev/test set now.

---

> > ### Comment · Reviewer_MNad · 2024-11-25
> >
> > Thanks for the authors' response.
> >
> > For the transitivity motivation, I would suggest to depict more clearly on why the transitivity can be important under the context of dense retrieval.
> >
> > For the new augmented QA datasets, can you provide more details on the data preparation, e.g. the number of original passages for the train/dev/test split, whether there are any overlaps, and whether there are any filtering/postprocessing?

---

> > > ### Author Response · Authors · 2024-11-27
> > >
> > > Thank you for your suggestion!
> > >
> > > **Regarding transitivity:**
> > > We have redefined our notions of 'transitivity' and 'non-transitivity' for the cosine function and Hoyer sparsity, providing a mathematical proof in Appendix D. The reason why 'transitivity'/'non-transitivity' matters for the contradiction retrieval task is the existence of a specific hard case: 'A contradicts C, B contradicts C, but A doesn’t contradict B' (see our example in Table 8, where A = original, B = paraphrase, and C = contradiction). Therefore, we argue that 'transitive' metrics are inherently incapable of characterizing this scenario.
> > >
> > > Specifically, we present the following propositions:
> > >
> > > **Proposition D.1 ("non-transitivity" of hoyer sparsity)** There exist three vectors A, B, and C of dimensionality d, satisfying $1\le \|A\|_2, \|B\|_2, \|C\|_2\le 1+O(1/\sqrt{d})$, such that $ Hoyer(A,C) > 1-O(1/\sqrt{d}) $, $Hoyer(B,C)> 1-O(1/\sqrt{d})$, and $Hoyer(A,B) < O(1/\sqrt{d})$.
> > >
> > > Please refer to Appendix D for the proof.
> > >
> > > **Proposition D.2 (“transitivity” of cosine function)** Given three unit vectors A, B, and C, if $cos(A,C)\ge 1-O(\epsilon)$ and $cos(B,C)\ge 1-O(\epsilon)$, we have $cos(A,B)\ge 1-O(\epsilon)$
> > >
> > > Please refer to Appendix D for the proof.
> > >
> > > **Regarding details of the augmented QA datasets:**
> > > We understand your concern about data overlap. As suggested by reviewer 83EJ, we have verified that there is no overlap between the MSMARCO and HotpotQA datasets we created. Therefore, we believe the zero-shot experiment we conducted in Table 2 is sound. Below, we provide the number of unique passages for MSMARCO and HotpotQA:
> > >
> > > |Dataset|Train|dev|test|
> > > |---|---|---|---|
> > > |MSMARCO|59500|5950|44101|
> > > |HotpotQA|59074|5896|81673|
> > >
> > > We are happy to discuss further if there are any remaining concerns.

---

> > > > ### Author Response · Authors · 2024-12-02
> > > >
> > > > Dear reviewer,
> > > >
> > > > We believe that we have addressed most of your concerns. Please let us know if we can address any of your additional comments/questions in the remaining time.

---

### Author Response · Authors · 2024-12-04
**Rebuttal Summary**

Dear AC and all the reviewers,

Thank you for your helpful feedback, as well as your time and effort during the discussion session. We have carefully reviewed the reviewers' comments and addressed most of the major points. Based on their suggestions, we have added new experiments and updated the paper accordingly.

We highlight that the reviewers find our work:

- Important and Practical Problem
  - The task of contradiction retrieval is meaningful and has practical applications, such as scientific claim verification and counter-argument retrieval.
  - The paper identifies limitations of existing methods (e.g., similarity-based methods and cross-encoders) and proposes corresponding solutions.

- Novelty of the Method
  - The idea of sparse-aware sentence embeddings and the use of the Hoyer measure is novel and well-motivated.
  - The Hoyer sparsity measure is non-transitive, effectively distinguishing indirectly related but contradictory sentences.
  - There was no method for retrieving contradictions that only relied on sentence embeddings.

- Empirical Effectiveness
  - SPARSECL is empirically validated on diverse datasets, including Arguana and synthetic data from MSMARCO and HotpotQA.
  - Results demonstrate consistent improvements over baseline methods, showing generalizability when integrated into different LLM-based retrievers.
  - Ablation studies justify the design choices and highlight the importance of individual components.

- Efficiency
  - SPARSECL achieves strong performance while being computationally efficient compared to cross-encoder-based approaches.

---

> ### Author Response · Authors · 2024-12-04
>
> We briefly summarize the reviewer’s concerns and our response:
> - Statistical Significance Testing
>   - Feedback: Statistical significance tests were not conducted to validate performance improvements.
>   - Response: We have conducted two-tailed t-tests to confirm whether the observed improvements are statistically significant and added the p-values in the experimental results section.
>
> - Dataset Diversity
>   - Feedback: The experiments are limited to one real dataset (Arguana) and two synthetic datasets. Additional benchmarks like NLI datasets are recommended.
>   - Response: We have included experiments on the NLI benchmark dataset to validate the generalizability of SPARSECL to other domains.
>
> - Task Justification
>   - Feedback: The importance of contradiction retrieval as a standalone task needs better justification.
>   - Response: We demonstrated how contradiction retrieval can enhance downstream tasks, such as claim verification and dataset cleaning, by providing concrete examples and results. (See section 4.4 and Appendix A)
>
> - Baseline Comparisons
>   - Feedback: The lack of strong baselines, including models adapted for contradiction retrieval, weakens the claims.
>   - Response: We added baseline models with context prefixes to compare against SPARSECL and demonstrate the necessity of the sparsity-based metric. (See Table 5)
>
> - Generalization and Scalability
>   - Feedback: The tuning of hyperparameters, such as the scalar in the scoring function, may limit generalization.
>   - Response: We explored testing with fixed hyperparameters and reported results on unseen datasets to validate robustness.
>
> - Data Contamination Concerns
>   - Feedback: The potential overlap between MSMARCO and HotpotQA may compromise generalization claims.
>   - Response: We conducted data-cleaning steps to ensure no contamination and ran the experiments with strict train/test splits.
>
> - Qualitative Analysis
>   - Feedback: The role of sparsity loss lacks detailed analysis and examples.
>   - Response: We added qualitative examples and visualizations showing how sparsity-based embeddings capture contradictions effectively. (See Appendix F)
>
> - Synthesis Bias
>   - Feedback: The synthetic datasets may favor the proposed method due to oversimplified contradictions.
>   - Response: We provide two reasons: First, we inject more paraphrases into the corpus to confuse the similarity search, which causes the standard similarity search to perform poorly in this case. Please refer to our dataset construction and result analysis (lines 359-364) in Section 4.2. Second, our synthetic HotpotQA and MSMARCO datasets contain more balanced paraphrases and contradiction pairs, providing 88k (original sentence, positive example, negative example) tuples for fine-tuning. For Arguana, we were only able to gather 14k such tuples for fine-tuning. Also, we have conducted experiments on the NLI benchmark dataset to validate the generalizability of SPARSECL to other domains.

---

### Meta-Review · Area_Chair_ABE7 · 2024-12-18

**Metareview:**

This paper proposes a novel retrieval method for retrieving contradicting passages for specific queries. Different from retrieving similar or relevant passages corresponding to a query, contradiction retrieval works in the opposite way. Very few methods have been proposed for this task, and traditional contrastive fine-tuning has been shown less effective for contradiction identification. To address this issue, this paper proposes to use a sparsity function, namely Hoyer sparsity, to contrastively train the embeddings to encourage the contradicting passages only differ in sparse dimensions. By combining both traditional cosine similarity with sparsity metric, the method achieves the best performances across one realistic dataset and two synthetically generated datasets.

Strengths:
- The task of contradiction retrieval is underexplored. This paper proposes an interesting metric relying on sparsity function to capture the nuances of contradicting pairs, which works well for this task.
- The proposed strategy is simple yet effective.
- Extensive experiments are conducted over one realistic and two synthetic datasets, validating the advantage of the method.

Weaknesses:
- The scope of this paper is a bit limited. The method is evaluated on one realistic dataset. It could be more interesting and impactful if the method could be applied and demonstrated to improve other downstream tasks, besides contradiction identification (since there is only one dataset available). There is also a lack of baseline.
- The authors use two additional synthetically generated datasets using GPT-4. While this could be useful, it is not shown whether the generated data is reliable, i.e., the contradicting pairs are indeed contradiction. Further discussions on the generated datasets could be given.
- The motivation of using weighted sum over cosine similarity and sparsity is not clearly given, as well as the motivation of incorporating cosine similarity. The scalar weight $\alpha$ largely depends on a specific dataset or domain, limiting its generalizability.
- The associations between this task and NLI and possible experimental comparisons could be added.

**Additional Comments On Reviewer Discussion:**

- Reviewers raised concerns on the scope, baseline models and benchmark datasets. While the authors added a few baselines suggested by the reviewer, the scope of this method in downstream tasks and the limitation of existing evaluation dataset are still a concern.
- Reviewers asked questions regarding the effect of this method on NLI datasets. The authors have conducted addtional experiments addressing the reviewers' concern.
- Additional questions regarding the choice of $alpha$ are raised. This could be limiting the generalizability and stability of this approach. Further, the motivation of using sparsity function and also combining with cosine similarity is lacking. The authors have provided more evidence and discussions to address these concerns.

---

### Decision · Program_Chairs · 2025-01-22

Reject